# Development of an Orchard Mowing and Sweeping Device Based on an ADAMS–EDEM Simulation

Shuai Shen [1,2], Yichuan He [1,2,3,*], Zhihui Tang [4], Yameng Dai [4], Yu Wang [1,2] and Jiaxin Ma [1,2]

1 College of Mechanical and Electronic Engineering, Tarim University, Alar 843300, China; 107572222@stumail.taru.edu.cn (S.S.); 10757222267@stumail.taru.edu.cn (Y.W.); 10757222228@stumail.taru.edu.cn (J.M.)
2 Agricultural Engineering Key Laboratory, Ministry of Higher Education of Xinjiang Uygur Autonomous Region, Tarim University, Alar 843300, China
3 College of Technology, Huazhong Agricultural University, Wuhan 430070, China
4 Mechanical Equipment Research Institute, Xinjiang Academy of Agricultural Reclamation Sciences, Shihezi 842000, China; xjtzh0701@sina.com (Z.T.); daiym832000@163.com (Y.D.)
* Correspondence: heyc@taru.edu.cn; Tel.: +86-198-8295-8652

**Abstract:** In the context of cutting grass in orchards, the practice of leaving cut weeds in the orchard rows hinders the decomposition of the weeds and the absorption of nutrients by the fruit trees. To address this issue, a grass-cutting machine with an integrated sweeping disc was designed to remove weeds from orchard rows and sweep them to the roots of the trees to promote their absorption of nutrients. A coupled simulation platform was established using multi-body dynamics ADAMS and the discrete element method EDEM. The weed-shedding and sweeping device was dynamically analyzed through an ADAMS–EDEM collaborative simulation that enabled the use of a second-order regression orthogonal rotation experiment and response surface methodology. The optimal parameters for the cutting tools, cutter shaft speed, and the number of cutting tools included 23 cutting tools arranged in a single helical pattern for the cutting device, a cutter shaft speed of 728 rpm, and claw-shaped blades as the cutting tools. A prototype machine was built based on the optimized parameters and tested in the field. The results indicated that, when there were 250 m² of weeds, the cutting rate reached 92.96%. The machine was highly maneuverable, and the average remaining weed height in the orchard was 110 mm, which met the national standards and local agricultural requirements. The new orchard grass-cutting and sweeping device meets the technical demands of orchard grass operations in the Xinjiang region of China.

**Keywords:** agricultural machinery; orchard management; mowing device; sweeping device





## 1. Introduction

China is a major nation for fruit production, and its area dedicated to the cultivation of orchards has been steadily growing each year. The development of smart agriculture has spurred the rapid growth of intelligent orchards and drives orchard management toward mechanization and informatization [1]. Weeds in orchards are a critical factor that affects the growth of fruit trees. In particular, the presence of large woody weeds can lead to competition for water, depletion of nutrients, infestation with pests and diseases, and light interference, which can severely impact the yield and quality of orchards [2,3]. Researchers have found that the technique of "living mulch" enables weeds to coexist with fruit trees. This not only has no adverse impact on the yield of fruit trees but can even enhance the production and quality of their fruit. The practice of living mulch contributes to increased biodiversity and organic matter inputs, reduced evaporation losses, and the prevention of soil erosion. It serves as a crucial means of regulating ecosystem services in orchards [4,5]. Consequently, grass cutting is a vital task in orchards, and mechanical grass-cutting machines are utilized for this purpose in the context of living

mulch technology [6,7]. Existing grass-cutting machines are predominantly designed for large-scale grass harvesting in pastures, which renders them unsuitable for use in orchards. Because the existing equipment has constraints, there is relatively little mechanization for cutting grass in orchards. The development of grass-cutting machines suitable for orchard environments holds significant importance in advancing living mulch technology and increasing orchard yields [8–10].

Previous research suggests significant advantages in orchard cultivation with the adoption of sod culture technology. Chen Jun's investigation found that natural sod treatment lowered mid-layer air temperature, increased air humidity, reduced light intensity, and enhanced leaf dry matter content, contributing to higher apple tree yields and income [11]. Hu Zhu Mei's research revealed that sod treatment regulated orchard soil surface and near-ground temperature [12]. Wu Hongmin's study indicated that sod culture effectively moderated soil and surface temperatures, positively influencing fruit tree growth [13]. Liu Haodong's analysis showed that maintaining a 6–8 cm stubble height enhanced grass yield and quality, emphasizing the importance of preventing excessive nutrient competition with fruit trees [14]. To further promote sod culture technology, the use of a grass cutter is essential. It trims grass, reduces nutrient competition, and ensures an optimal weed height, effectively implementing sod culture technology. In addressing the challenges of the complex terrain in mountainous citrus orchards, Mapanyu [15] designed a citrus orchard mimicry mower cutter that utilized a flexible blade design with an obstacle avoidance functionality to reduce the damage to the blades from obstacles. Wu Bei [16] aimed to solve the problem of harvesting alfalfa (*Medicago sativa* L.) and developed a small self-propelled mower that utilized a mimicry device. However, it is only suitable for large pastures and for use in orchards. Yang Tian [17] designed an articulated steering orchard mower that only focused on changes in steering to address the poor maneuverability and inconvenience to personnel of existing orchard mowers in hilly areas, with limited research on other aspects of mowing. Zhu Lu [18] optimized the folding mechanism and hydraulic mimicry system of a mower to enhance its operational capability on a complex terrain, but, overall, the equipment created is relatively large and only suitable for extensive forest use. Hu Wenxiu [19] designed a rolling mower, which, compared to rotary mowers, is more effective in trimming and can trim lawns more cleanly. Although this research is not fully developed, it has significant implications for improving the techniques of mowing grass. Previous studies have primarily focused on terrain challenges, operational difficulties, and mowing effects, among others, and the developed mowers are only suitable for use in some pastures and lawns. In China and other countries, weed control technology in orchards has slowly developed, and there has been little research that specifically addresses inter-row weeding.

In response to the limited research on orchard weeding today and to better understand the impact of weeds on the growth of fruit trees, increase their yield, and improve the income of farmers, this study aimed to develop a mower. It used a combination of theory, simulation, and experiments to design a sweeping-type orchard mower with the intent of providing equipment support and a research point of reference to improve the development of technology for mowing grass in orchards.

## 2. Materials and Methods

### 2.1. Overall Structure and Working Principle

#### 2.1.1. Overall Structure

As shown in Figure 1, the grass-cutting and sweeping device primarily consisted of support wheels, sweeping discs, a sweeping disc motor, sweeping disc connecting brackets, an obstacle avoidance hydraulic cylinder, an obstacle avoidance cutting blade hydraulic motor, an obstacle avoidance disc, inter-row cutting blades, in-row cutting blades, a hydraulic pump, a hydraulic reservoir, and a chemical tank, among other key components. The tractor transmitted power to the gearbox, which, in turn, provided power to the hydraulic pump and drove the hydraulic components of the device. A portion of the power was allocated to the inter-row cutting blade shaft section. The overall mechanism is

illustrated in Figure 1. For detailed three-dimensional drawings, see Appendix A for three dimensional drawings.

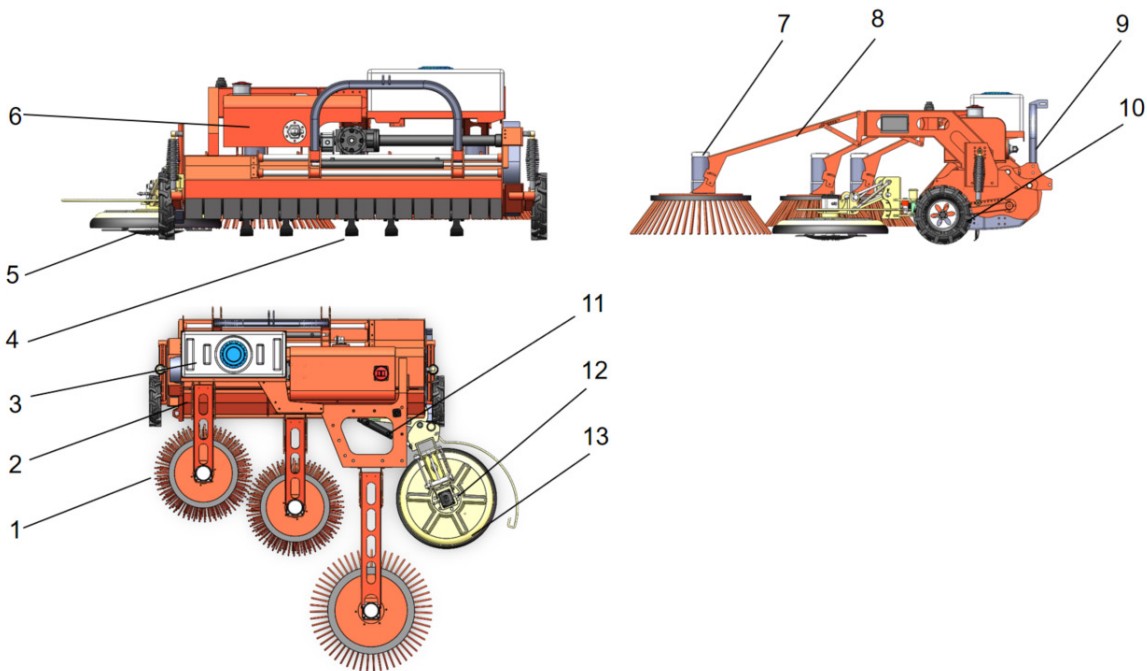

**Figure 1.** Orchard obstacle avoidance mowing and sweeping device, with the following components: 1. Sweeping disc; 2. mower casing; 3. water tank; 4. hammer claw blade; 5. inter-row tossing blade; 6. hydraulic oil tank; 7. reduction motor; 8. sweeping disc connecting frame; 9. connecting frame; 10. support wheel; 11. obstacle avoidance hydraulic cylinder; 12. hydraulic motor; and 13. obstacle avoidance disc.

The sweeping device was located behind the inter-row cutting blades. After the inter-row cutting blades divided the weeds, they would fall to the ground. The three sweeping discs rotated to clear the cut weeds closer to the tree trunk. The collected weeds were sprayed with a bioagent to accelerate their decomposition, which turned them into nutrients for the fruit trees.

### 2.1.2. Working Principle

The grass-cutting blade was powered by the tractor, which drove its rotation. Simultaneously, the obstacle-avoiding grass-cutting blade was equipped with a hydraulic motor to propel its rotation. The cleaning discs required lower speed and torque, and, thus, we opted for a reduction motor to perform this task. When the machine was operated, the rotating grass-cutting blade severed the weeds, while the cleaning discs rotated counterclockwise. This gradually cleared the weeds in-between the rows and pushed them toward the vicinity of the tree trunks. Through the rotation of the three cleaning discs, the severed weeds were gathered at the base of the tree trunks. Subsequently, the machine's water pump sprayed a biotic agent, which promoted the decomposition of the weed piles, thus making it easier for the fruit trees to absorb nutrients. The power transmission principle of the machine is shown in Figure 2.

During the operation of the orchard inter-row obstacle avoidance grass-cutting and sweeping machine, the following obstacle avoidance steps were utilized, as shown in Figure 3a. In this phase, the obstacle avoidance disc moved toward the position of the tree trunk and then approached it. The obstacle avoidance process was initiated, and the obstacle avoidance rod sensed the position of the tree trunk by contact. When it made contact with the tree trunk, the obstacle avoidance rod transmitted pressure to the signal receiver, which then issued a command to make the electromagnetic directional valve

change direction. Subsequently, the hydraulic cylinder retracted to move the obstacle avoidance disc away from the tree trunk, as shown in Figure 3b. During this process, the obstacle avoidance disc encountered resistance from the tree trunk, which triggered the retraction of the hydraulic cylinder to safely bypass the obstacle. Once the obstacle avoidance disc had completely cleared the tree trunk during its forward movement, as shown in Figure 3c, the hydraulic cylinder extended, and the obstacle avoidance disc returned to the inter-row position. This prepared it to cut the grass. The obstacle avoidance process was completed, as shown in Figure 3d. At this stage, the obstacle avoidance mechanism had successfully guided the obstacle avoidance disc around the tree trunk, which enabled the machine to continue the grass cutting within rows. This obstacle avoidance process ensured that the machine could safely navigate around obstacles, such as tree trunks, during its operation, preventing damage to the machine or the trees. The design of this mechanism enabled the grass to be cut while avoiding orchard obstacles, and the sweeping machine efficiently cut grass and cleared weeds in the orchard while maintaining the integrity of the trees.

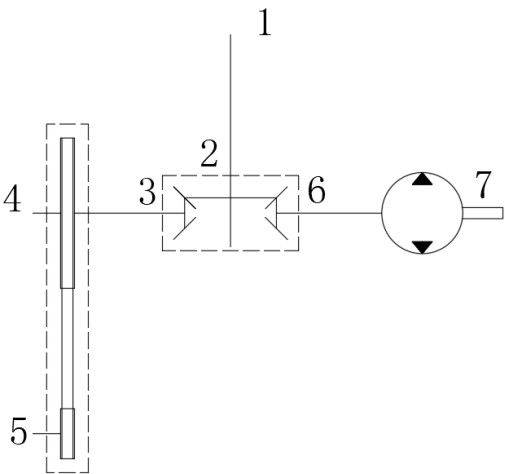

**Figure 2.** The schematic diagram of power transmission for the machine, consisting of the following components: 1. input shaft; 2. input bevel gear I; 3. output bevel gear; 4. large pulley; 5. small pulley; 6. output bevel gear II; and 7. hydraulic motor.

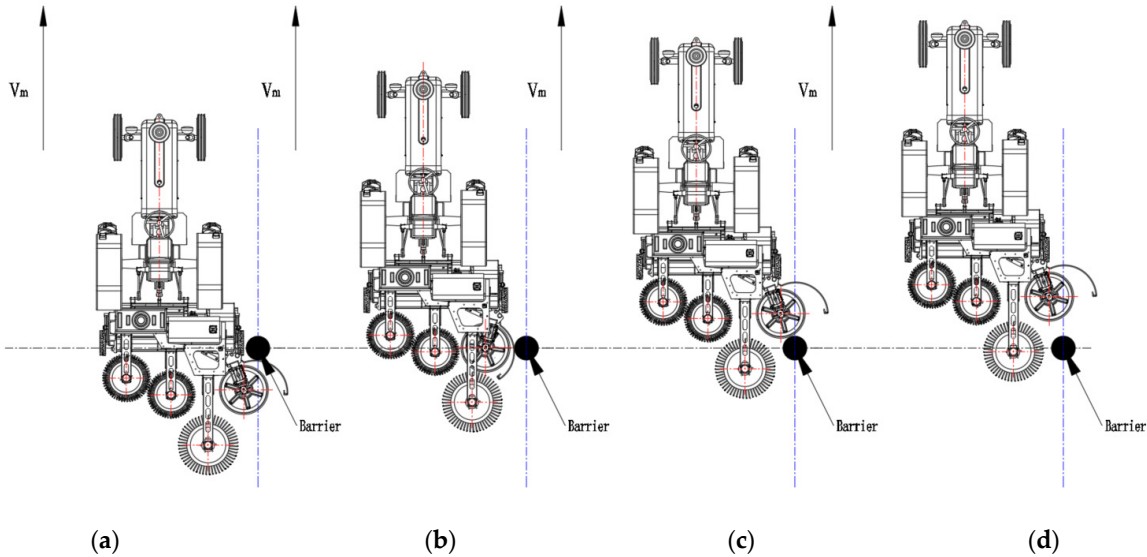

**Figure 3.** Working principle of a smart mower for an orchard. (**a**) The identification-of-a-fruit-tree stage; (**b**) the hydraulic-pressure-system-working stage; (**c**) the machine-avoids-the-fruit-tree stage; and (**d**) the obstacle avoidance end stage.

### 2.2. Design of Key Components

#### 2.2.1. Inter-Row Mowing Components

To be able to cut the grass by avoiding obstacles inter-rows, two distinct forms of cutting tools were utilized. These included the inter-row cutting blade and the around-the-tree cutting blade. After the inter-row weeds were severed, the cleaning disc relocated them closer to the tree trunk, which enhanced the ability of the fruit trees to absorb nutrients from the remnants of the weeds. The around-the-tree cutting blade was designed with a rotating mechanism, which enabled it to avoid the tree trunk while simultaneously clearing the surrounding weeds.

First, among these components, there was the hammer-claw-type blade, with a rotating radius (R) of 105 mm, a blade handle 40 mm wide (L), and a blade edge 70 mm wide (H). Owing to its substantial mass and outward center of gravity, the hammer-claw-type blade exhibited a significant moment of inertia during rotation, which resulted in an excellent crushing performance. This tool was highly adaptable and effectively pulverized both soft-stemmed and hard-stemmed crops. Furthermore, it resisted damage when it encountered hard objects, such as rocks, during its use. Thus, it avoids the issues that are typically associated with L-shaped blades during fieldwork [20,21]. The schematic diagrams of the hammer-claw blade and the cutting blade roller are shown in Figures 4 and 5, respectively.

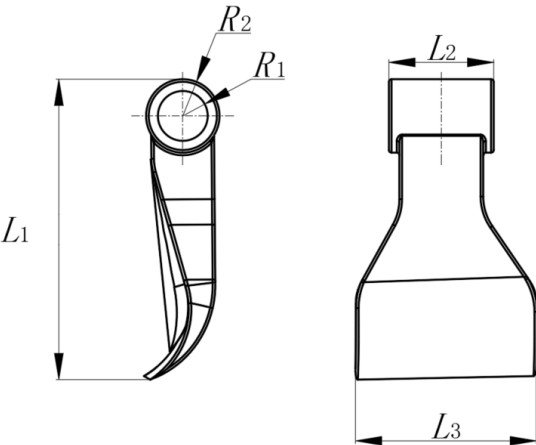

**Figure 4.** Schematic diagram of the cutter structure. Note: $L_1$ is the total length of the hammer claw knife, in mm; $L_2$ is the width of the shank of the hammer claw knife, in mm; $L_3$ is the width of the blade of the hammer claw knife, in mm; $R_1$ is the radius of the shank shaft of the hammer claw knife, in mm; and $R_2$ is the radius of the mounting hole of the hammer claw knife, in mm.

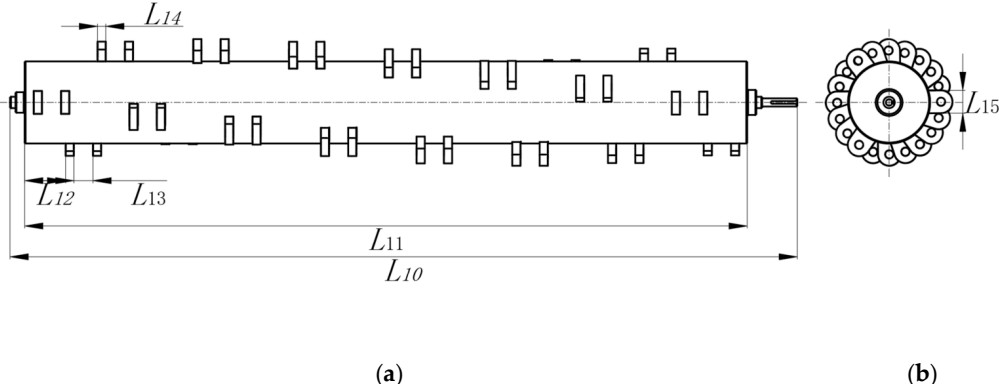

(**a**)            (**b**)

**Figure 5.** Schematic structure of the orchard inter-row mowing cutter shafts. (**a**) Main view; (**b**) side view. Note: $L_{11}$ is the length of the blade shaft, in mm; $L_{12}$ is the width of the blade holder, in mm; $L_{13}$ is the distance from the left blade holder to the axis edge, in mm; $L_{14}$ is the distance between two blade holders, in mm; $L_{15}$ is the thickness of the blade holder, in mm; and $L_{16}$ is the width of the blade holder, in mm.

### 2.2.2. Inter-Plant Cutting Components

The throwing blade was designed in a Z-shaped configuration. The side edge structure of the throwing blade significantly impacted its cutting effectiveness on the weeds. A smaller blade angle cut much more effectively. However, excessively small angles may reduce its lifespan. Therefore, a blade angle of 30° was selected [22]. The bending width of the throwing blade was designed to be 5 cm. Factoring in the stubble height [23] and previous research [22], a blade that was 120 mm long was deemed suitable. Given the materials available in the market for grass-cutting blades, the preliminary choice for the throwing blade material was 65 Mn that was 5 mm thick, and it was subjected to a quenching treatment. The Z-shaped blade structure is illustrated in Figure 6.

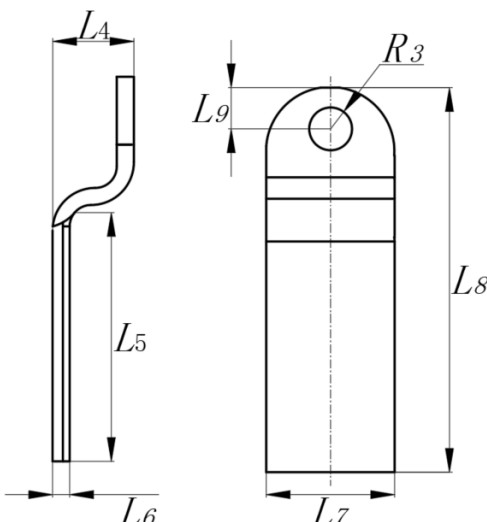

**Figure 6.** Schematic diagram of the structure of the dump knife. Note: $L_4$ is the total height of the Z-shaped blade, in mm; $L_5$ is the length of the Z-shaped blade edge, in mm; $L_6$ is the thickness of the blade edge, in mm; $L_7$ is the width of the blade, in mm; $L_8$ is the total length of the blade, in mm; $L_9$ is the distance from the mounting hole to the blade handle, in mm; and $R_3$ is the radius of the mounting hole, in mm.

### 2.2.3. Sweeping Device

The mounting plate for the cleaning apparatus was cut to the appropriate size from Q235 steel. After the cutting process, holes were drilled to accommodate the installation of rigid nylon strips in alignment with the mounting holes. This ensured that the bottom of the cleaning disc remained level. Subsequently, the entire machine was assembled and tested. The schematic diagram of the cleaning disc structure is shown in Figure 7.

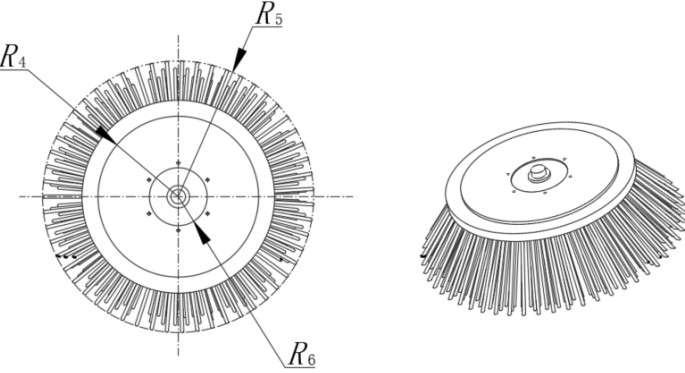

**Figure 7.** Schematic diagram of the structure of the sweeping disk. Note: $R_4$ is the radius of the fixed plate for the sweeping disc, in mm; $R_5$ is the radius of the maximum sweeping area for the sweeping disc, in mm; $R_6$ is the radius of the mounting flange for the sweeping disc, in mm.

### 2.3. Simulation Test of Grass-Cutting Device

2.3.1. Simulation Modeling in EDEM

(1) Simulation model of weed stalks

"EDEM" is a simulation software that employs the Discrete Element Method (DEM) to model particle system behavior. The DEM is a numerical simulation technique extensively used in granular flow, granular dynamics, and particle interactions. In the EDEM 2022 simulation experiments, a Hertz-Mindlin with bonding contact model was used to simulate the mechanical properties of the weed stems [24,25]. In this study, the physical parameters of the weeds in the orchard were tested and analyzed during their mature stage [26]. As shown in Table 1, the results identified average values of 912 mm in height and 6 mm in diameter.

**Table 1.** Parameters of the particle model.

| Name | Position X (m) | Position Y (m) | Position Z (m) | Rotation Y (deg) |
|------|----------------|----------------|----------------|------------------|
| particle 0 | 0 | 0 | 0 | 90 |
| particle 1 | 0 | 0 | 0.045 | 90 |
| particle 2 | 0 | 0 | 0.095 | 90 |
| particle 3 | 0 | 0 | 0.145 | 90 |
| particle 4 | 0 | 0 | 0.195 | 90 |
| particle 5 | 0 | 0 | 0.245 | 90 |
| particle 6 | 0 | 0 | 0.295 | 90 |
| particle 7 | 0 | 0 | 0.345 | 90 |
| particle 8 | 0 | 0 | 0.395 | 90 |
| particle 9 | −0.03 | 0 | 0.449 | 45 |
| particle 10 | 0.03 | 0 | 0.449 | −45 |

The spatial coordinates for 10 cylindrical weed particles were established in EDEM, as indicated in Table 1. Subsequently, the final model for the 10 cylindrical weed particles was created, as shown in Figure 8. These particle models were generated within the particle factory according to the principles of normal distribution [27–30].

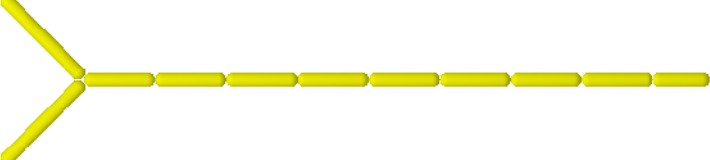

**Figure 8.** Model of weed particles.

(2) Setting of the property parameters of the test materials

A discrete element model was established using the EDEM software for the grass-cutting device, the weed stems, and the cleaning apparatus. The grass-cutting device was made of cast steel, and the cleaning disc's bristles were constructed from nylon. As an example, alfalfa was chosen as the weed, with material properties which included Poisson's ratio, density, and elastic modulus, as shown in Table 3 [25]. The model for the weed stems [26] was characterized by a particle radius of 5 mm, a contact radius of 0.55 mm, a normal stiffness of $1.5 \times 10^{10}$ N m$^3$, a tangential stiffness of $1 \times 10^{10}$ Pa per unit area, critical normal stress of $5 \times 10^8$ Pa, critical tangential stress of $5 \times 10^8$ Pa, and a bond radius of 0.55 mm [26], with material property parameters as shown in Table 2.

**Table 2.** Properties parameters of the material.

| Material | Poisson's Ratio | Densities (kg/m³) | Modulus of Elasticity (Pa) |
|---|---|---|---|
| Mowing device | 0.3 | 7800 | $1.72 \times 10^{11}$ |
| Sweeping device | 0.3 | 1150 | $1.4 \times 10^{11}$ |
| Clover stalks | 0.4 | 227 | $1 \times 10^{6}$ |

(3)    Determination of the contact parameters

In the simulation experiment, contacts were established between three sets of parameters [31–33], including alfalfa and alfalfa, cast steel and alfalfa, and nylon and alfalfa. The material contact parameters were compiled from the literature [25] and are shown in Table 3.

**Table 3.** Discrete element model contact parameters.

| Contact Model | Coefficient of Static Friction | Coefficient of Rolling Friction | Coefficient of Restitution |
|---|---|---|---|
| Lucerne–lucerne | 0.3 | 0.3 | 0.01 |
| cast steel–lucerne | 0.3 | 0.3 | 0.01 |
| Nylon–Alfalfa | 0.3 | 0.2 | 0.03 |

(4)    Earth trough simulation modeling

In the soil trough simulation, a structure with dimensions of 2500 mm × 5000 mm × 10 mm was constructed. It contained 400 alfalfa stems, and the discrete elements were modeled using the Hertz-Mindlin with bonding model, as shown in Figure 9.

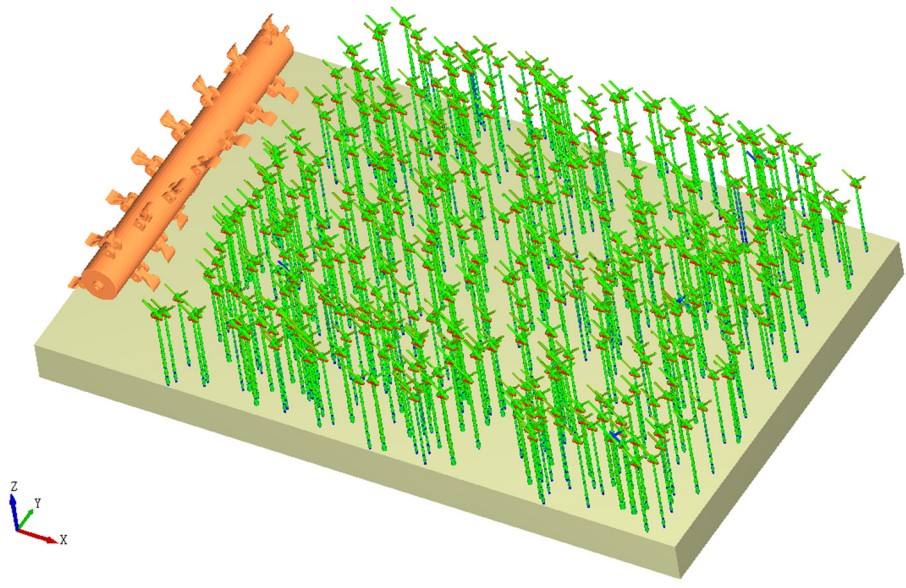

**Figure 9.** Discrete element model of the inter-row mowing device.

2.3.2. A Motion Simulation Model Was Constructed in Adams

"Adams" is a simulation software for multibody dynamics, specifically designed to model the dynamic behavior of mechanical systems. Developed by the "MSC Software Corporation", it is formally known as "MSC Adams". This software offers a comprehensive suite of tools dedicated to analyzing, optimizing, and simulating the dynamic performance of diverse mechanical systems. The three-dimensional (3D) model of the grass-cutting mechanism was simplified as shown in Figure 10. The simulation runs were conducted using the ADAMS 2020 software to ensure a proper configuration. Subsequently, the model construction in ADAMS was completed, which enabled further testing while maintaining the stability of the machine.

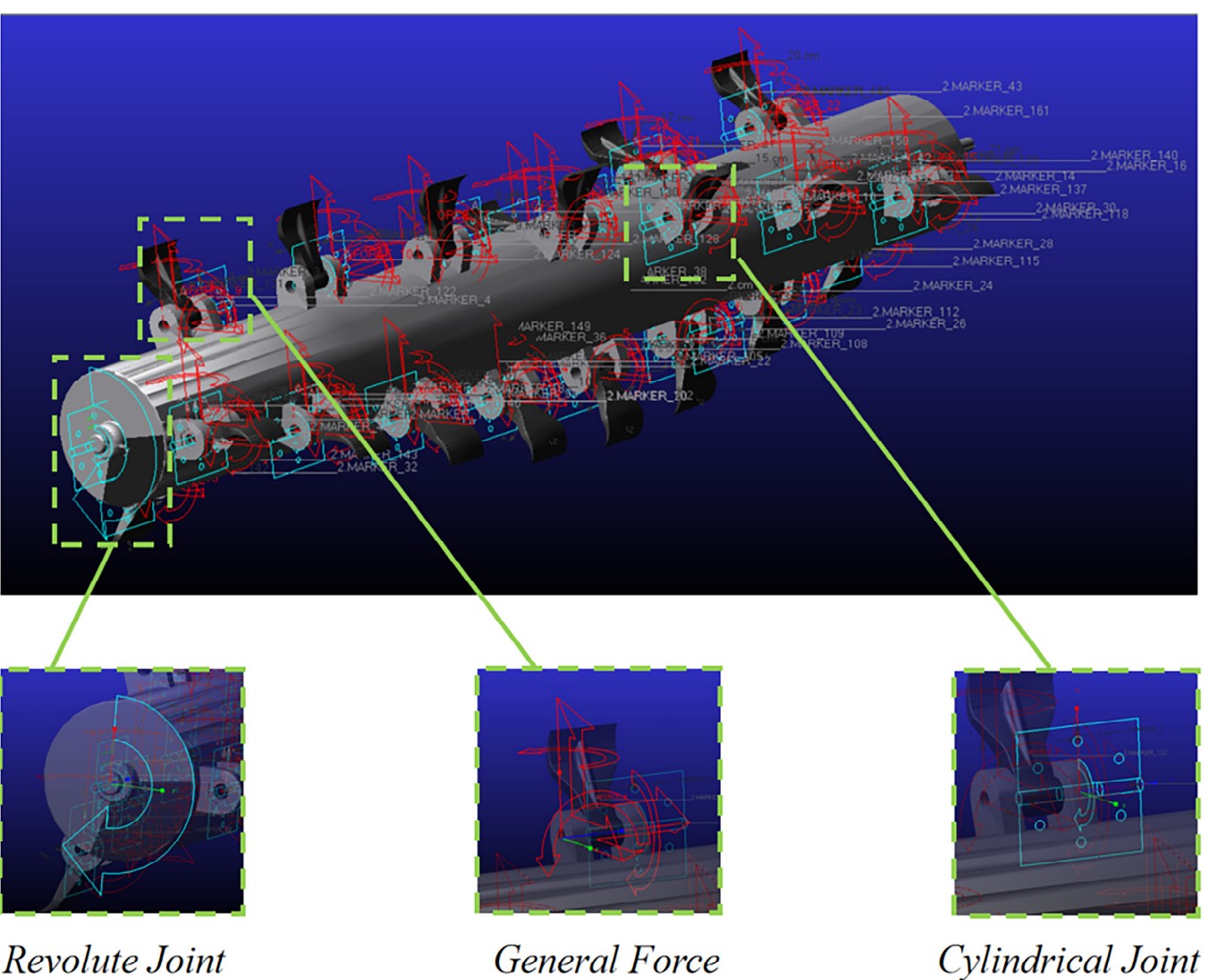

**Figure 10.** ADAMS simulation model.

### 2.3.3. ADAMS–EDEM Coupled Analytical Test

A combined ADAMS and EDEM simulation model was established to analyze the grass-cutting mechanism and verify the rationality of the hammer-claw blade cutting. Multiple test scenarios were established to validate the optimal parameters, including forward speeds which ranged from 0.6 to 1.4 m s$^{-1}$, cutting blade roller speeds between 540 and 850 rpm, and a variable number of cutting blades, from 17 to 29. The EDEM software was utilized to simulate the motion trajectory of the cutting blades and the fracture of the weed stems during its operation [34–37]. Several key time points were identified during the process of selecting the model, as shown in Figure 11. There was contact between the cutting blade and the weed stems at 0.18 s. The weed stem began to bend under the influence of the cutting blade at 0.3 s, and it began to fracture at 0.5 s. All the weed stems that were initially in contact with the cutting blades were completely fractured at 0.7 s. This marked the completion of one full rotation of the cutting blade roller in the grass-cutting process. The effectiveness of a weed stem fracture was influenced by the speed and forward velocity of the cutting blade.

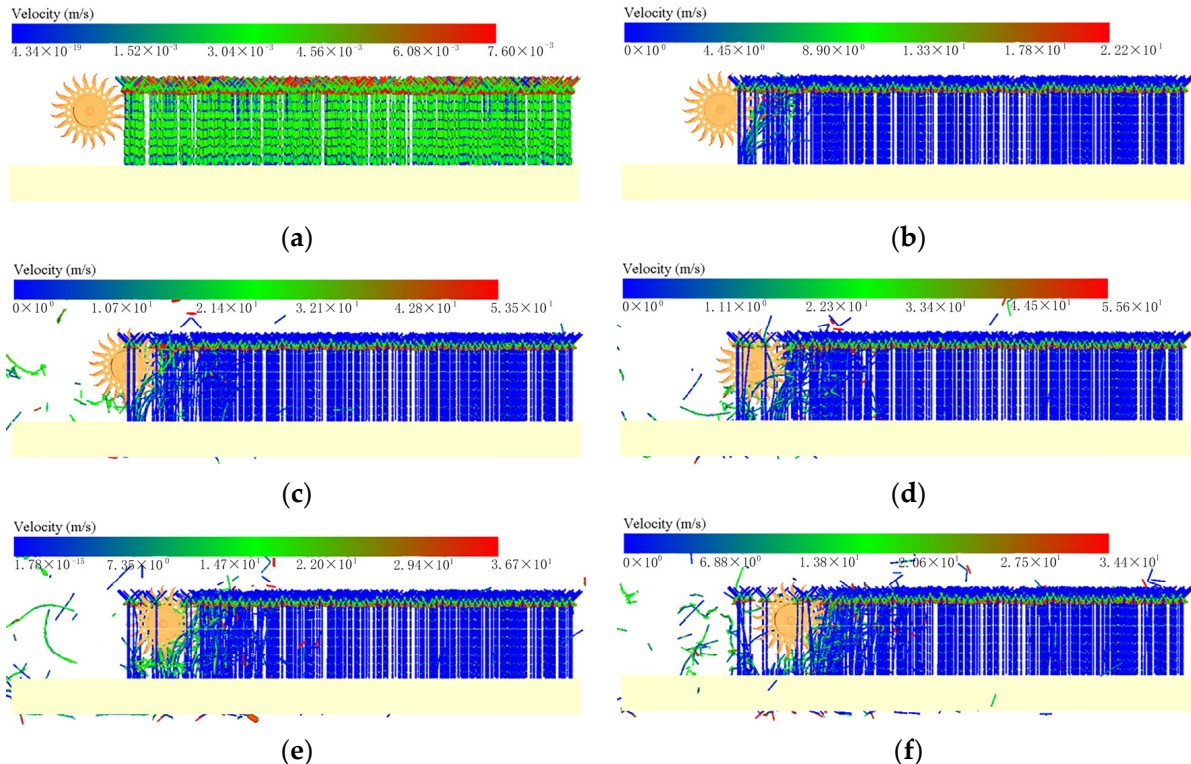

**Figure 11.** Schematic of the mowing simulation. Note: (**a**) T = 0.1800021448 s; (**b**) T = 0.2000014839 s; (**c**) T = 0.3000008771 s; (**d**) T = 0.400002703 s; (**e**) T = 0.5000023611 s; and (**f**) T = 0.6000017543 s.

The discrete element model for the alfalfa stems was based on the Hertz-Mindlin contact model with a bonding contact model. It utilized bonding links to cause the particles to adhere. These bonding links could withstand some level of external force before breaking [38,39]. When the cutting blade severed the weed stems, the number and power of the broken bonding links in the discrete element model for the alfalfa stems were influenced by the shear stress. Upon completion of the simulation analysis, post-processing was conducted by exporting data to scrutinize parameters such as the count of bond failures, spindle torque, and traction resistance [26]. The power is determined as shown in Equation (1) [27].

$$P = \frac{nM}{9550} + Fv_m \tag{1}$$

where $P$ represents the power (kW); $M$ is the torque of the cutting blade shaft (N·m); $n$ denotes the rotational speed (rpm), and $F$ signifies the traction resistance (N).

### 2.3.4. Simulation Analysis of the Sweeping Parts

To verify that the cleaning apparatus effectively collected the weeds, the shredded stems on the ground were swept away after the grass had been cut. The EDEM software was used to simulate the operational process. The simulation results demonstrated that the cleaning apparatus effectively swept and collected the weeds. The process of cleaning operation is shown in Figure 12.

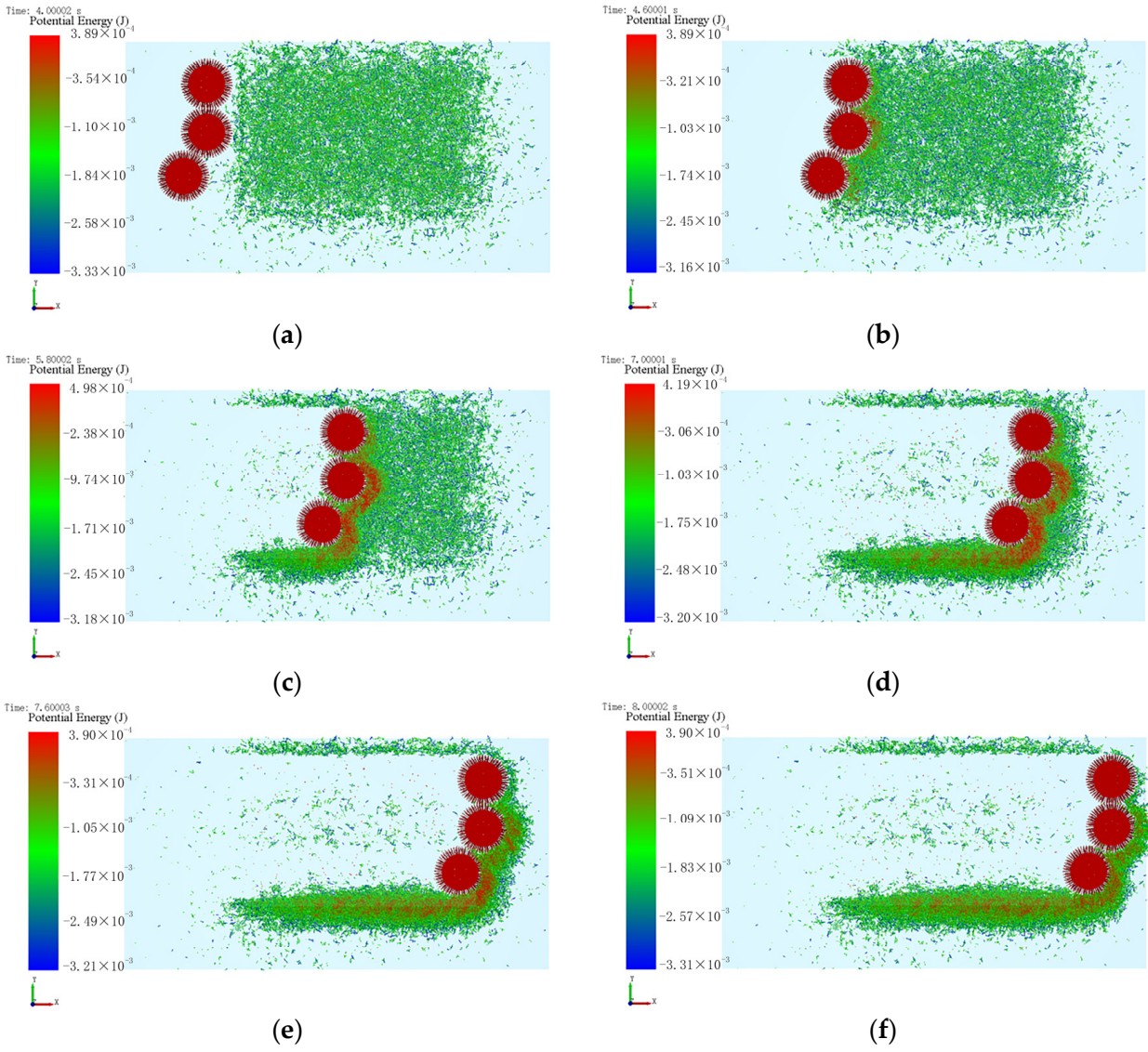

**Figure 12.** Simulation of the cleaning process. Note: (**a**) T = 4.00002 s; (**b**) T = 4.60001 s; (**c**) T = 5.80002 s; (**d**) T = 7.00003 s; (**e**) T = 7.60002 s; and (**f**) T = 8.00002 s.

## 3. Results and Discussion

### 3.1. Experimental Scheme and Results

The Design-Expert 13 software and the response surface methodology of a Box–Behnken Design (BBD) were utilized to design the experiments. They focused on an analysis of the coupling simulation results, particularly the number of broken bonding links and the power. The main influencing factors that were selected for this analysis were the forward velocity, the cutting blade rotational speed, and the number of cutting blades. The set ranges for these factors were as follows: the forward velocity ranged from 0.6 to 1.4 m s$^{-1}$, the rotational speed of the cutting blade from 540 to 850 rpm, and the number of cutting blades from 17 to 29, as depicted in the factor level table shown in Table 4. In order to reduce the number of experiments, a Box–Behnken Design (BBD) was chosen for this study. A total of seventeen sets of experiments were conducted, and each set was repeated three times. The final average values were selected. The experiments considered the interaction of all the factors, with the number of broken bonding links and the power consumption as the key metrics. The results are summarized in Table 5.

**Table 4.** Experimental factors and levels.

| Level | Test Factors | | |
| --- | --- | --- | --- |
| | A: Forward Speed (m/s) | B: Cutter Speed (r/min) | C: Cutter Quantity |
| −1 | 1.4 | 850 | 29 |
| 0 | 1 | 695 | 23 |
| 1 | 0.6 | 540 | 17 |

**Table 5.** Experimental program and results.

| Serial Number | Test Factors | | | Test Indicators | |
| --- | --- | --- | --- | --- | --- |
| | A m/s | B r/min | C | $Y_1$ | $Y_2$ kW |
| 1 | 1 | 540 | 29 | 2789 | 3.23 |
| 2 | 1.4 | 850 | 23 | 3515 | 3.04 |
| 3 | 0.6 | 695 | 29 | 3302 | 3.42 |
| 4 | 1 | 540 | 17 | 3827 | 3.74 |
| 5 | 1 | 695 | 23 | 3503 | 3.01 |
| 6 | 1 | 695 | 23 | 3501 | 3.05 |
| 7 | 0.6 | 695 | 17 | 2473 | 2.78 |
| 8 | 1.4 | 540 | 23 | 3170 | 3.32 |
| 9 | 1.4 | 695 | 29 | 3512 | 3.62 |
| 10 | 1 | 695 | 23 | 3511 | 3.05 |
| 11 | 1 | 695 | 23 | 3505 | 2.99 |
| 12 | 1 | 850 | 17 | 3171 | 3.42 |
| 13 | 1 | 695 | 23 | 3491 | 3.03 |
| 14 | 0.6 | 540 | 23 | 2403 | 2.57 |
| 15 | 1.4 | 695 | 17 | 3021 | 3.11 |
| 16 | 0.6 | 850 | 23 | 3287 | 3.31 |
| 17 | 1 | 850 | 29 | 3827 | 3.74 |

*3.2. Analysis of the Experimental Results*

The data in Table 6 were subjected to a multivariate regression analysis using Design-Expert 13, which led to the development of a second-order polynomial response surface regression model with three independent variables. These included the number of broken bonding links, the power consumption, the forward velocity (*A*), the rotational speed (*B*), and the number of cutting blades (*C*). The mathematical model was expressed as follows:

$$\begin{cases} Y_1 = -10853.89 + 7357.24A + 18.25B + 258.09C - 4.6AB - 14.38AC + 0.11BC \\ \quad -1545.94A^2 - 0.01B^2 - 5.88C^2 \\ Y_2 = 10.17 + 3.45A - 0.01B - 0.48C - 0.001AB - 0.04AC + 0.00002BC \\ \quad -0.51A^2 + 5.9B^2 + 0.008C^2 \end{cases} \quad (2)$$

**Table 6.** Analysis of variance table of the test results.

| Source of Variance | Number of Bond Breaks | | | | | |
| --- | --- | --- | --- | --- | --- | --- |
| | Sum of Squares | Freedom | Mean Square | F-Value | *p*-Value | Significance |
| Model | $3.135 \times 10^6$ | 9 | $3.483 \times 10^5$ | 18.52 | 0.0004 | ** |
| A | $6.903 \times 10^5$ | 1 | $6.903 \times 10^5$ | 36.70 | 0.0005 | ** |
| B | $6.659 \times 10^5$ | 1 | $6.659 \times 10^5$ | 35.40 | 0.0006 | ** |
| C | $6.261 \times 10^5$ | 1 | $6.261 \times 10^5$ | 33.29 | 0.0007 | ** |
| AB | $3.26 \times 10^5$ | 1 | $3.260 \times 10^5$ | 17.33 | 0.0042 | ** |
| AC | 4761.00 | 1 | 4761.00 | 0.2531 | 0.6303 | |
| BC | 38,809.00 | 1 | 38,809.00 | 2.06 | 0.1940 | |
| A2 | $2.576 \times 10^5$ | 1 | $2.576 \times 10^5$ | 13.70 | 0.0076 | ** |
| B2 | $2.545 \times 10^5$ | 1 | $2.545 \times 10^5$ | 13.53 | 0.0079 | ** |

**Table 6.** *Cont.*

| Source of Variance | Number of Bond Breaks | | | | | |
| | Sum of Squares | Freedom | Mean Square | F-Value | *p*-Value | Significance |
|---|---|---|---|---|---|---|
| C2 | $1.890 \times 10^5$ | 1 | $1.890 \times 10^5$ | 10.05 | 0.0157 | * |
| Residuals | $1.317 \times 10^5$ | 7 | 18,808.83 | | | |
| Failure to fit | $1.053 \times 10^5$ | 3 | 35,112.33 | 5.34 | 0.0698 | |
| Error | 26,324.8 | 4 | 6581.20 | | | |
| Total | $3.267 \times 10^6$ | 16 | | | | |

| Source of Variance | Power kW | | | | | |
| | Sum of Squares | Freedom | Mean Square | F-Value | *p*-Value | Significance |
|---|---|---|---|---|---|---|
| Model | 1.45 | 9 | 0.1616 | 4.63 | 0.0027 | ** |
| A | 0.5305 | 1 | 0.5305 | 15.19 | 0.0059 | ** |
| B | 0.1250 | 1 | 0.1250 | 3.58 | 0.1003 | |
| C | 0.0545 | 1 | 0.0545 | 1.56 | 0.2519 | |
| AB | 0.0182 | 1 | 0.0182 | 0.5221 | 0.4934 | |
| AC | 0.0462 | 1 | 0.0462 | 1.32 | 0.2876 | |
| BC | 0.1722 | 1 | 0.1722 | 4.93 | 0.0618 | |
| A2 | 0.0281 | 1 | 0.0281 | 0.8061 | 0.3991 | |
| B2 | 0.0864 | 1 | 0.0864 | 2.48 | 0.1597 | |
| C2 | 0.3872 | 1 | 0.3872 | 11.09 | 0.0126 | * |
| Residuals | 0.2444 | 7 | 0.0349 | | | |
| Failure to fit | 0.2033 | 3 | 0.0678 | 6.59 | 0.050 | |
| Error | 0.0411 | 4 | 0.0103 | | | |
| Total | 1.70 | 16 | | | | |

Note: Significant when $0.01 \leq p < 0.05$, expressed as *; extremely significant when $p < 0.01$, expressed as **.

A variance analysis of the model was conducted, and the results are shown in Table 6. The *p*-values for the models of the number of broken bonding links and power consumption were <0.01, while the *p*-values for the mismatched terms were >0.05. The model determination coefficients ($R^2$) for the number of broken bonding links and the power consumption were 0.96 and 0.86, respectively. These values indicated that the optimized regression model was statistically significant, fit well, and was reliable.

*3.3. Fitting Response Surface Method*

Design-Expert 13 was used to generate the response surface graphs, as shown in Figures 13 and 14, to further investigate the patterns of influence of the test factors and their interactions with the test values [40].

Figure 13 illustrates a response surface analysis of the interaction factors that affected the number of broken bonding links ($Y_1$). Increasing the forward velocity (A) and the rotational speed of the cutting blades (B) could increase the number of broken bonding links. However, when the number of broken bonding links reached its peak, it decreased, as the forward velocity (A) and the cutting blade rotational speed (B) increased.

When maintaining a constant number of cutting tools, Figure 13a illustrates the interactive influence of the feed rate and cutter speed on the quantity of bond failures. Clearly, at a sustained feed rate, an increase in the cutter speed is associated with a heightened number of bond failures. Similarly, when the cutter speed remains constant, an elevation in the feed rate corresponds to a greater quantity of bond failures.

In Figure 13b, the interactive effect of the feed rate and the number of cutting tools on the quantity of bond failures is presented, assuming a constant cutter speed. It is evident that, with a consistent feed rate, an increase in the number of cutting tools is linked to a higher number of bond failures. Conversely, when the number of cutting tools is fixed, an escalation in the feed rate leads to an increased quantity of bond failures.

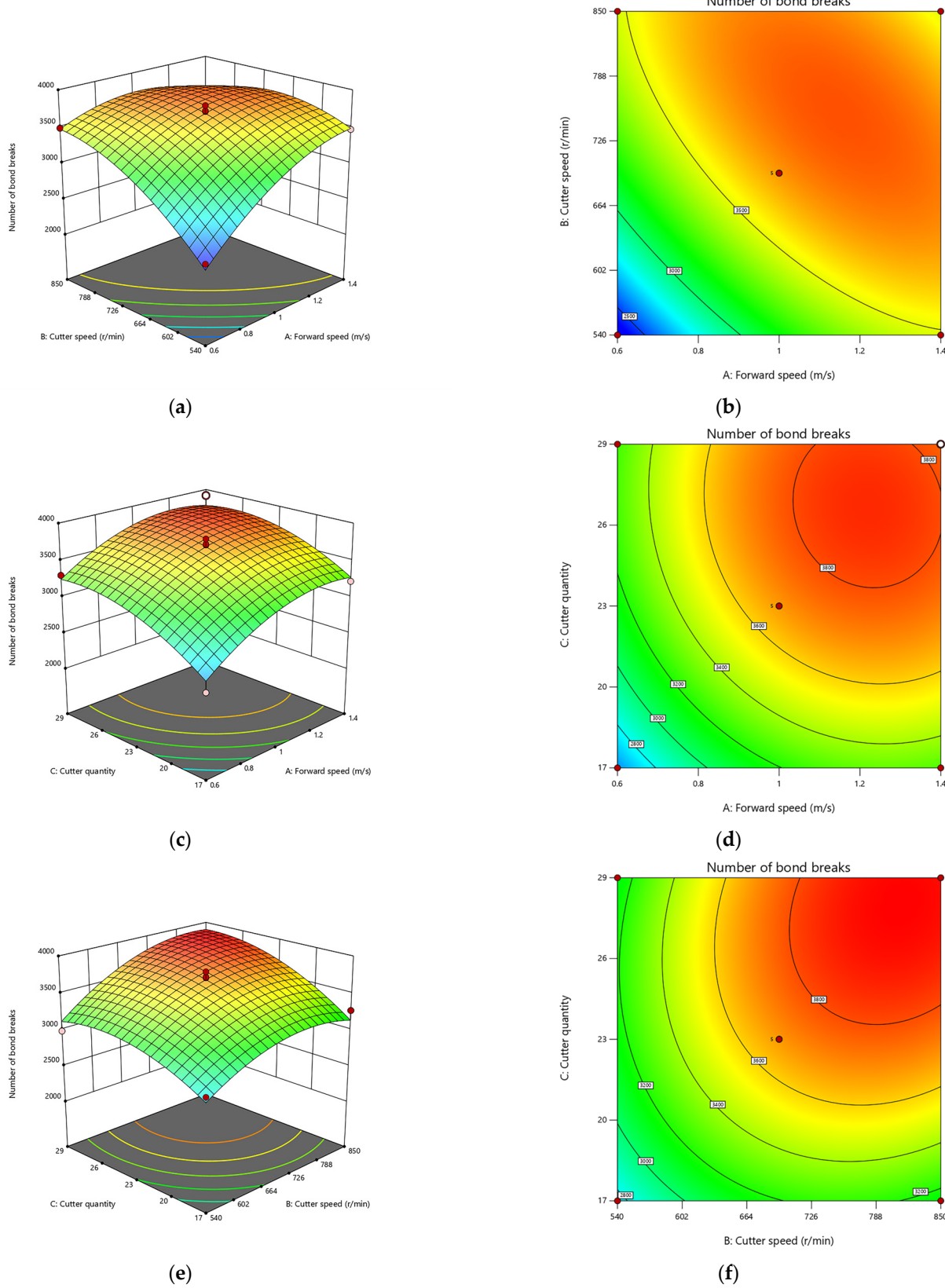

**Figure 13.** Response surface analysis of the interaction factors affecting the number of broken bonding links. (**a**) Response surface plot of the effect of the interaction between the machine's forward speed and the rotational speed of the cutter on the number of bond breaks. (**b**) Contour plot of the effect of the interaction between the machine's forward speed and the rotational speed of the

cutter on the number of bond breaks. (**c**) Response surface plot of the effect of the interaction between the machine's forward speed and the number of cutters on the number of bond breaks. (**d**) Response surface plot of the effect of the interaction between the machine's contour plot of the effect of the forward speed of the machine and the number of cutters on the interaction with the number of bond breaks. (**e**) Response surface plot of the effect of the rotational speed of the cutter and the number of cutters on the interaction with the number of bond breaks. (**f**) Contour plot of the effect of the rotational speed of the cutter and the number of cutters on the interaction with the number of bond breaks.

Figure 13c illustrates the interactive effect of the feed rate and the number of cutting tools on the quantity of bond failures, assuming a constant cutter speed. It is apparent that, with a consistent feed rate, an increase in the number of cutting tools results in a higher incidence of bond failures. Similarly, when the number of cutting tools is constant, an upturn in the cutter speed leads to an increase in the quantity of bond failures.

When the number of cutting tools is constant, Figure 14a illustrates the interactive effect of the feed rate and cutter speed on the power. It is observed that, with a constant cutter speed, an increase in the feed rate has a negligible impact on the power. Similarly, at a fixed feed rate, an increase in the cutter speed initially results in an increase in the power, followed by a decrease.

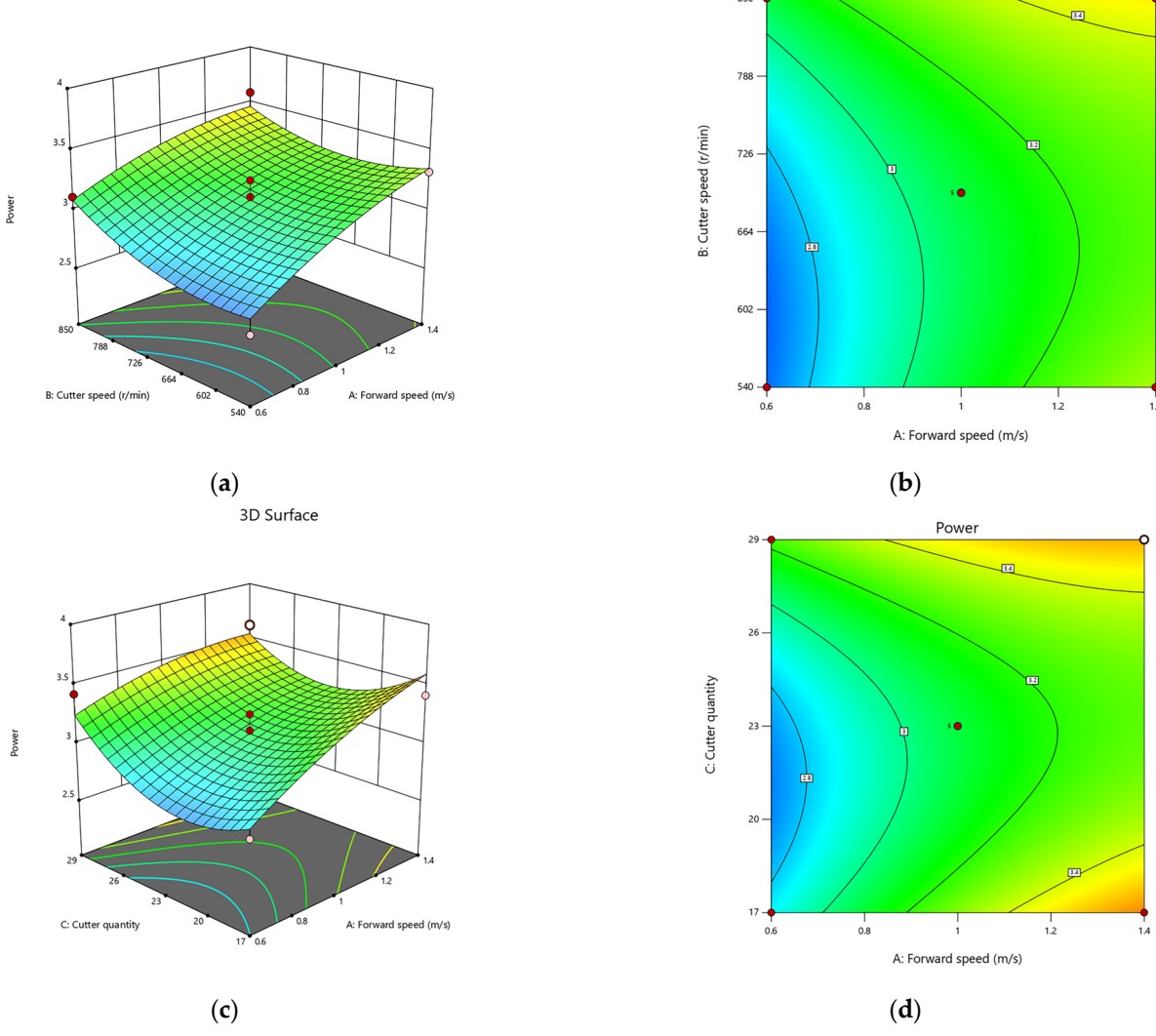

(**a**)

(**b**)

(**c**)

(**d**)

**Figure 14.** *Cont.*

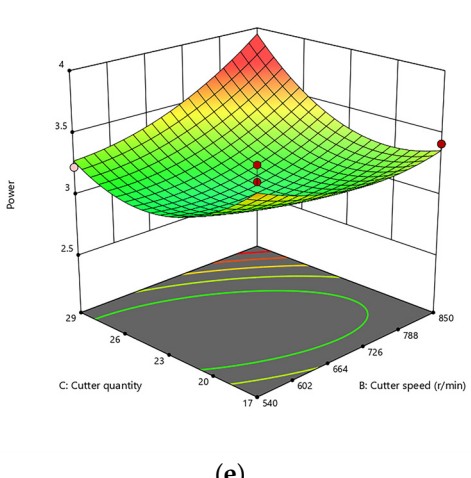

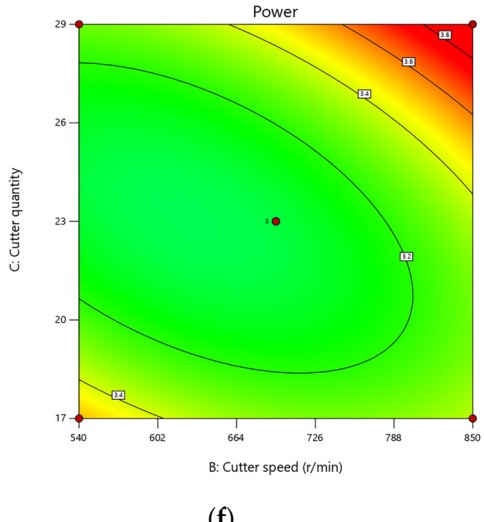

(**e**)                                                                                                                                (**f**)

**Figure 14.** Response surface analysis of the interaction factors affecting the operating power. (**a**) Response surface plots of the effect of the machine's forward speed and the cutter rotational speed on the interaction with the implemented power. (**b**) Contour plots of the effect of the machine's forward speed and the cutter rotational speed on the interaction with the implemented power. (**c**) Response surface plots of the effect of the machine's forward speed and the number of cutters on the interaction with the implemented power. (**d**) Contour plot of the effect of the machine's forward speed and the number of cutters on the interaction with the implemented power. (**e**) Response surface plot of the effect of the rotational speed of the cutters and the number of cutters on the interaction with the implemented power. (**f**) Contour plot of the effect of the rotational speed of the cutters and the number of cutters on the interaction with the implemented power.

Figure 14b demonstrates the interactive effect of the feed rate and the number of cutting tools on the power when the cutter speed is constant. It is evident that, with a consistent feed rate, an increase in the number of cutting tools initially leads to a decrease in power, followed by an increase. Conversely, when the number of cutting tools is fixed, an increase in the feed rate has a minor effect on the power.

In Figure 14c, the interactive effect of the feed rate and the number of cutting tools on the power is depicted when the cutter speed is constant. It is observed that, with a constant feed rate, an increase in the number of cutting tools initially leads to a decrease in power, followed by an increase. Similarly, when the number of cutting tools is fixed, an increase in the cutter speed initially results in an increase in the power, followed by a decrease.

### 3.4. Optimization Model Analysis and Laboratory Test Verification

Based on the results of the regression model which utilized the test metrics described above and the operating conditions of the grass-cutting apparatus, constraints were established for each test factor in Design-Expert 13. The objective function was defined as shown in Equation (3), and the optimal combination of parameters was determined to be a forward velocity of 0.79 m s$^{-1}$, a cutting blade rotational speed of 728.254 rpm, and 23 cutting blades. There were 3550 broken bonding links and a power consumption of 2.97 kW.

$$\begin{cases} minY_1 = f_1(A,B,C) \\ minY_2 = f_2(A,B,C) \\ S.T. \begin{cases} Y_1 \leq 3912 \\ Y_2 \leq 3.74 \\ 0.6\text{m/s} \leq A \leq 1.4\text{m/s} \\ 540\text{rpm} \leq B \leq 850\text{rpm} \\ 17 \leq C \leq 29 \end{cases} \end{cases} \tag{3}$$

The optimized parameters were rounded to the nearest values and subsequently verified to validate the reliability of the model's predictions. The forward velocity was set at 0.8 m s$^{-1}$; the cutting blade rotational speed was set at 728 rpm, and 23 cutting blades were used. Testing was conducted three times, and the results were averaged. The results from the experimental validation closely matched those of the optimization analysis. Thus, these parameters were established as the optimal operating conditions for the grass-cutting machine.

## 4. Field Test

The prototype was manufactured based on the combination of optimal parameters, and field experiments were conducted to verify whether the machine met the operational requirements.

### 4.1. Test Conditions and Equipment

A performance test of grass cutting was conducted in August 2023 at the Jiu Tuan Xiangli Demonstration Orchard in Alar City, Xinjiang Uygur Autonomous Region, China, to assess whether the mower met the Rotating Lawn Mower Standard (GB/T 10938-2008) [41]. The test area primarily grew dogtail grass (*Cynosurus echinatus* L.) and alfalfa, with a density of approximately 220 plants per m$^2$. The grass height ranged from 200 to 600 mm, and the grass stem diameter was approximately 3–6 mm. The tractor used for the operation was a New Dongfang 604 model, and the grass was cut at 0.99 m s$^{-1}$. The orchard's grass-cutting and sweeping apparatus was connected to the tractor through a three-point suspension and utilized a universal joint to transmit power between the gearbox input shaft and the power output shaft of the tractor. Orchard mowing and sweeping device detailed drawing see Appendix A physical drawing.

### 4.2. Test Method

4.2.1. Measurement of the Stubble Height

Two points to measure the stubble height were selected at intervals of 5 m along each working pass, and the stubble height was measured using a steel ruler. The average stubble height for each pass was determined [38].

$$\bar{h} = \frac{\sum h}{n} \tag{4}$$

where $\sum h$ $S$ represents the total sum of $n$ measurements of stubble height for each pass in cm, and $n$ is the number of measurements of stubble height taken during each pass.

For each test area, the stubble height was measured at regular intervals along the cutting width while determining the cutting width. The stubble height was measured consecutively along the test area, as follows:

$$u_j = \left(1 - \frac{s_j}{\bar{h}}\right) \times 100\% \tag{5}$$

where $u_j$ represents the stubble height stability coefficient (%), and $s_j$ is the standard deviation of the stubble height.

4.2.2. Determination of the Leakage Rate

In each measurement area, nine testing areas that were 1 m long and 1.5 m wide were selected at equal intervals along the forward direction of the grass-cutting machine. Within the respective testing areas, the number of missed cuts and the total number of cuts were determined. Subsequently, the missed cut rate was calculated as follows [42]:

$$\lambda_j = \frac{n_j}{N_j} \times 100\% \tag{6}$$

where $\lambda_j$ represents the missed cut rate (%); $n_j$ is the number of missed cuts, and $N_j$ is the total number of cuts.

### 4.3. Test Results and Analysis

4.3.1. Measurement of Stubble Height

As shown in Figure 15, the experiments were conducted in the standardized fruit orchard demonstration area. The stubble height was measured at 5 m intervals with a total of nine measurements. As shown in Table 7, the average stubble height was 0.11 m, which indicated that the grass-cutting apparatus effectively preserved the stubble at this specific height.

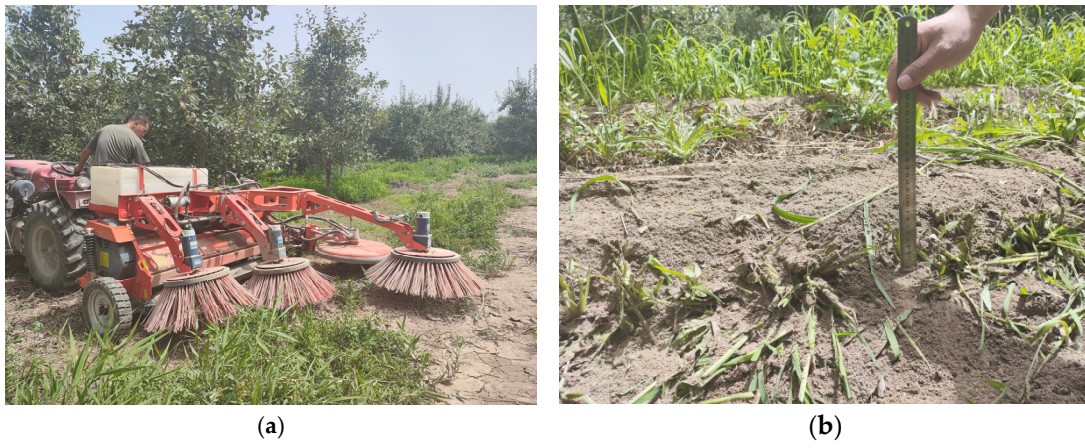

(**a**)　　　　　　　　　　　　　　　　　　　　(**b**)

**Figure 15.** Orchard test. Note: (**a**) orchard mowing site; (**b**) stubble height.

**Table 7.** Stubble height test results.

| Serial Number | Mowing Position/m | Stubble Height/m | Average Height/m |
|:---:|:---:|:---:|:---:|
| 1 | 20 | 0.12 | |
| 2 | 25 | 0.13 | |
| 3 | 30 | 0.09 | |
| 4 | 35 | 0.15 | |
| 5 | 40 | 0.08 | 0.11 |
| 6 | 45 | 0.12 | |
| 7 | 50 | 0.09 | |
| 8 | 55 | 0.11 | |
| 9 | 60 | 0.10 | |

4.3.2. Determination of the Leakage Rate

The test results for the uncut area and the missed cut rate within the nine 1.5 m² testing areas are shown in Table 8. The average missed cut rate between the rows was 7.04%.

**Table 8.** Leakage rate test results.

| Serial Number | Measurement of Total Area/m² | Uncut Area/m² | Undercutting Rate/% | Average Leakage Rate/% |
|:---:|:---:|:---:|:---:|:---:|
| 1 | 1.5 | 0.14 | 9.3 | |
| 2 | 1.5 | 0.09 | 6.0 | |
| 3 | 1.5 | 0.10 | 6.7 | |
| 4 | 1.5 | 0.09 | 6.0 | |
| 5 | 1.5 | 0.12 | 8.0 | 7.04 |
| 6 | 1.5 | 0.12 | 8.0 | |
| 7 | 1.5 | 0.10 | 6.7 | |
| 8 | 1.5 | 0.09 | 6.0 | |
| 9 | 1.5 | 0.10 | 6.7 | |

## 5. Conclusions

An analysis of the current state of research indicates that the development of machinery for weed control in orchards is not advanced. To address the gap in handling weeds after cutting and navigate through inter-row obstacles, a cutting and sweeping device for grass in orchards was developed. This was accomplished in conjunction with the promotion and application of grass technology in local orchards in Xinjiang. The feasibility of the machine in accomplishing its intended tasks was comprehensively assessed through the use of simulation software, field testing, and other methods.

The point of this study was to address the current research gaps in weed management post orchard cutting and inter-row obstacle avoidance whilst grass cutting. Thus, it designed a novel grass-cutting and sweeping device. The main components were determined by analyzing key components such as the inter-row cutting blades, the intra-row cutting blades, and the sweeping device.

An ADAMS–EDEM coupled model and a discrete element model for the weed stems and the grass-cutting and clearing device have been established. A Box–Behnken response surface optimization experimental method was used to analyze the influence of the forward speed, the cutting blade rotation speed, and the number of cutting blades on the number of bonding keys and the power. Regression models were developed, and their interactions showed that the verification experiments closely aligned with the results of the optimization analysis. This demonstrated the reliability of the regression model that had been optimized to the parameters.

The field tests indicated that, when the grass-cutting and clearing device operated at $0.8 \text{ m s}^{-1}$, the average post-cut stubble height was 110 mm. Thus, this device met the requirements for agronomic operations. The inter-row missed cut rate was 7.04%. The equipment performed well and met the operational requirements of modern grass management techniques in Xinjiang, as well as the local agronomic requirements. In comparing the optimized parameters with the pre-optimization data, it is evident that using the optimized results for the operations results in a stable stubble height and reduced missed cutting rates. This leads to a more effective completion of orchard weed removal tasks.

With the rapid development of the Information Age, agriculture has also undergone significant changes. In order to better free-up human labor and improve operational efficiency, the next goal for this piece of equipment is for it to be fully mechanized and automated. Farmers should be able to remotely control the equipment to remove weeds.

**Author Contributions:** Resources, Y.H.; data curation, Z.T.; writing—original draft preparation, S.S.; writing—review and editing, Y.W.; visualization, Y.D.; supervision, Y.H.; project administration, J.M. All authors have read and agreed to the published version of the manuscript.

**Funding:** This research was financially supported by the Bingtuan Science and Technology Program (2021AA005 and 2021AA0050302).

**Institutional Review Board Statement:** Not applicable.

**Informed Consent Statement:** Informed consent was obtained from all the subjects involved in the study.

**Data Availability Statement:** The data presented in this study are available on request from the corresponding author.

**Acknowledgments:** The authors thank Yichuan He from Tarim University for his thesis supervision. The authors are grateful to the anonymous reviewers for their comments.

**Conflicts of Interest:** The authors declare no conflict of interest.

## Appendix A

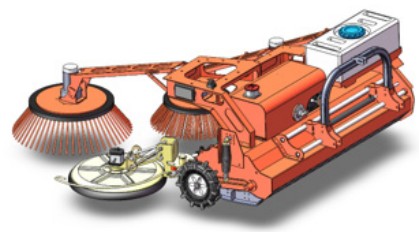 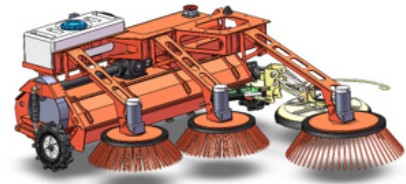 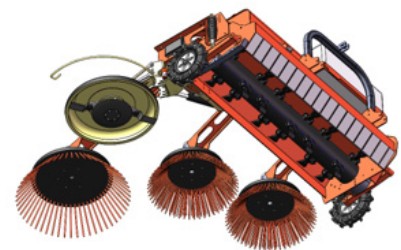

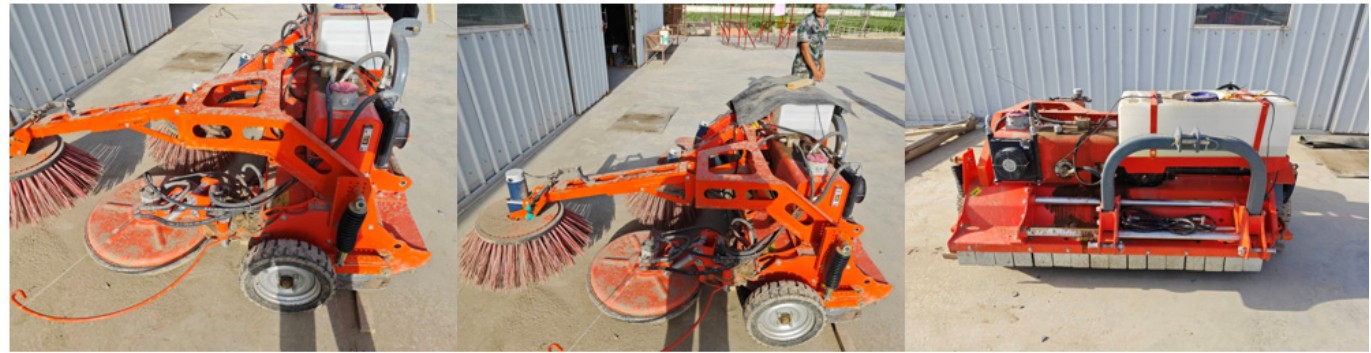

**Figure A1.** The three-dimensional drawing and physical drawings of the Orchard Mowing and Sweeping Device.

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
