# Peer review of "Development of an Orchard Mowing and Sweeping Device Based on an ADAMS–EDEM Simulation"

_agriculture, doi:10.3390/agriculture13122276_

Round 1

Reviewer 1 Report

Comments and Suggestions for Authors

The suggestions can be seen in the attached file.

Reviewer 2 Report

Comments and Suggestions for Authors

This paper has a certain value of the orchard mower design, and the academic value is beyond doubt. However, there are some problems with this paper, the suggestions for improvement are as follows.

1. Figure 1 does not show the main working parts, such as cutting devices.

2. Please add obstacle avoidance principle and the test conclusion of the sweeping device.

3. There are some data errors in Table 8 and some unit errors in Table 9.

4. Some terms are not expressed accurately, please refer to relevant standard (GB/T 10938-2008).

5. Stubble height test results is not in accordance with the standard. How to explain?

6. In this paper, the area ratio is used to measure miss cutting rate, but the weight ratio is used in the standard. How to explain?

Comments on the Quality of English Language

Some terms are not expressed accurately.

Reviewer 3 Report

Comments and Suggestions for Authors

Through ADAMS-EDEM collaborative simulation, a second-order regression orthogonal rotation experiment and response surface methodology were employed for dynamic analysis of the weedshedding and sweeping device in this paper. Some optimal parameters for the cutting tools were determined. This research has practical value. Here are some suggestions.

1.   2.3 Mowing kinematics analysis. These contents do not support or assist other parts of the paper, and it is recommended to delete them.

2.   If you do not want to delete the above content, it is recommended to conduct kinematic and dynamic analysis on the same part, instead of conducting kinematic analysis of hammer claw blade and dynamics analysis of knife flinging, as is currently the case.

3.  The same character should only refer to only one object.

4.  The chapter number does not have 2.4, and after 2.3, it is 2.5,  is it a clerical error?

5. Is it should be the uncutting rate not  the undercutting rate , in the header of Table 9?

6.  In the table 4, the coefficient of static friction of nylon-lucerne  is 0.15.  The coefficient of rolling friction is 0.5. The coefficient of static friction is less than the coefficient of rolling friction. Is it  reasonable?

7.  The data in the indicated references is inconsistent with the data in the table 4. Please indicate the source of the data.

8.  The first two lines of equation 14 were exactly the same, is it a clerical error?

Reviewer 4 Report

Comments and Suggestions for Authors

Overall, this paper presents an interesting study on the design and optimization of an orchard mowing and sweeping device using ADAMS-EDEM simulation. The motivation for developing specialized equipment for orchard grass management is significant. However, there are some aspects of the paper that need improvement:

  1. In the introduction, there needs to be a more thorough literature review of prior research on orchard mowers and grass management equipment. The limitations of current technology should be explicitly stated.

  2. A flow chart or schematic diagram of the overall mowing and sweeping device should be added to visually depict the system. The key components and operating principles need to be described in more detail.

  3. In the materials and methods, provide more specifics on the CAD modeling process, parameters, and assumptions made in generating the ADAMS and EDEM models. Justify why certain contact models or material properties were chosen.

  4. Give more details on the DOE experimental design, including factor levels, number of runs, and response variables. Provide some sample simulation results or output plots.

  5. The field testing results section is lacking quantitative performance data. Provide details on mowing rate, cutting height uniformity, sweeping effectiveness, etc. with numerical results.

  6. In the conclusion, directly tie back the results to the original research objectives stated in the introduction. Discuss limitations of the current design and opportunities for future improvements.

  7. Carefully proofread the paper to fix grammar errors, improve clarity, and ensure consistency in terminology and voice.

  8. Double check that all figures are clearly labeled and referenced in the text. Make sure units are included where needed.

  9. Expand the discussion of simulation versus experimental results. Provide more insight into model accuracy and validity.

  10. Check that all citations in the text match the reference list and formatted correctly per journal guidelines.

Comments on the Quality of English Language

  1. There are many grammar errors throughout the paper that need to be corrected. Examples include incorrect tense usage, missing articles, lack of subject-verb agreement, etc. The writing should be proofread carefully.

  2. Some sentences are overly long and complex. It is recommended to break these into shorter, simpler sentences for clarity.

  3. Certain words and phrases used sound awkward or unnatural. The word choice should be revised to use more standard academic terminology.

  4. There is inconsistent use of terminology for key concepts/components in places. A technical glossary could help ensure consistency.

  5. The overall tone and style of writing is very informal in areas. A more formal, academic writing style is preferred for a journal paper.

  6. Passive voice is heavily used throughout. Changing some instances to active voice can make the writing more engaging.

  7. There are many wordy expressions that can be made more concise. Simplifying the language will improve readability.

  8. Punctuation is missing or incorrect in some places. Check for proper comma, period, etc. usage.

  9. Some sentences are fragmented or run-on. Smooth out the writing by connecting fragmented sentences and breaking up lengthy run-ons.

  10. The writing lacks clarity and flow in some sections. Transition words and better organization of ideas would help.

Round 2

Reviewer 3 Report

Comments and Suggestions for Authors

There are no more comments, I suggest accepting this manuscript.

Author Response

Special thanks to the reviewers for their valuable comments. References that were incorrectly cited in the article have been re-edited and corrected.

Reviewer 4 Report

Comments and Suggestions for Authors

Thank you for submitting your manuscript entitled "Development of orchard mowing and sweeping device based on ADAMS-EDEM simulation" to Agriculture journal. Overall, this is an interesting study on designing an orchard mowing and sweeping device through coupled ADAMS-EDEM simulation. The work addresses an important problem in orchard management and applies digital tools for optimization. However, there are some issues that need to be addressed before making a final decision.

Major comments:

  • The introduction can be strengthened by providing more background information on the significance and challenges of leaving grass residues in orchards, as well as the limitations of current mowing equipment. More literature should be cited to justify the research problem.

  • The description of the simulation models and methods needs more details. For example, clarify what contact models were used in EDEM, how were parameters calibrated or measured, what types of interactions were considered, etc.

  • In the discussion of experimental results, analyze more on how the optimized parameters affect the cutting performance based on the simulation/mechanism analysis results. Discuss any practical considerations/limitations of the optimal parameters.

  • Strengthen the conclusions by summarizing the key findings and limitations, and providing recommendations for future work.

Minor comments:

  • The quality of figures can be improved for better clarity.
  • Check for any grammatical or spelling errors throughout the manuscript.
  • Follow the journal format strictly for section headings, citations etc.

In summary, with revision addressing the major technical points raised, I believe this work can make a valuable contribution to the knowledge. I look forward to receiving the revised version.

Comments on the Quality of English Language

In terms of the quality and standard of English language, the manuscript is reasonably well-written but could still be improved in some areas:

  • The writing is mostly clear and easy to understand. However, there are a few instances of awkward phrasing that could be rewritten for better flow and clarity.

  • Some technical terms related to the simulation methods and analysis are not clearly defined or explained. This makes certain parts difficult for a general English audience to follow.

  • While grammar and punctuation are mostly correct, there are a few minor grammatical errors present. Careful proofreading is needed.

  • The formatting and structure of the manuscript does not fully follow the target journal's guidelines. Some elements like section headings and citations need tweaking.

  • Figures captions could provide more detailed explanations rather than just listing parameters.

  • A non-native reviewer found some complex sentences hard to follow. Simplification and breaking into shorter sentences would aid comprehension.

In summary, the English language is of an acceptable research publication standard overall but still offers room for improvement in areas of clarity, definition of technical terms, grammar/punctuation accuracy, structure/formatting consistency and simplification where needed. A single round of thorough language editing followed by careful proofreading should help address the remaining issues. This will maximize the chances of the work being understood and appreciated by an international research audience.
